# Mixtures of strategies underlie rodent behavior during reversal learning

**Nhat Minh Le**[1,2], **Murat Yildirim**[2,3], **Yizhi Wang**[2], **Hiroki Sugihara**[1,2], **Mehrdad Jazayeri**[1,4], **Mriganka Sur**[1,2]*

**1** Department of Brain and Cognitive Sciences, Massachusetts Institute of Technology, Cambridge, Massachusetts, United States of America, **2** Picower Institute for Learning and Memory, Massachusetts Institute of Technology, Cambridge, Massachusetts, United States of America, **3** Department of Neurosciences, Cleveland Clinic Lerner Research Institute, Cleveland, Ohio, United States of America, **4** McGovern Institute for Brain Research, Massachusetts Institute of Technology, Cambridge, Massachusetts, United States of America

\* msur@mit.edu

**Data Availability Statement:** Data for the study is available at https://doi.org/10.6084/m9.figshare. 22493308. Code used for data analysis and synthetic simulations are available at https://github. com/nhat-le/block-hmm-simulations. Code for

## Abstract

In reversal learning tasks, the behavior of humans and animals is often assumed to be uniform within single experimental sessions to facilitate data analysis and model fitting. However, behavior of agents can display substantial variability in single experimental sessions, as they execute different blocks of trials with different transition dynamics. Here, we observed that in a deterministic reversal learning task, mice display noisy and sub-optimal choice transitions even at the expert stages of learning. We investigated two sources of the sub-optimality in the behavior. First, we found that mice exhibit a high lapse rate during task execution, as they reverted to unrewarded directions after choice transitions. Second, we unexpectedly found that a majority of mice did not execute a uniform strategy, but rather mixed between several behavioral modes with different transition dynamics. We quantified the use of such mixtures with a state-space model, block Hidden Markov Model (block HMM), to dissociate the mixtures of dynamic choice transitions in individual blocks of trials. Additionally, we found that blockHMM transition modes in rodent behavior can be accounted for by two different types of behavioral algorithms, model-free or inference-based learning, that might be used to solve the task. Combining these approaches, we found that mice used a mixture of both exploratory, model-free strategies and deterministic, inference-based behavior in the task, explaining their overall noisy choice sequences. Together, our combined computational approach highlights intrinsic sources of noise in rodent reversal learning behavior and provides a richer description of behavior than conventional techniques, while uncovering the hidden states that underlie the block-by-block transitions.

## Author summary

Humans and animals can use diverse decision-making strategies to maximize rewards in uncertain environments, but previous studies have not investigated the use of multiple strategies that involve distinct latent switching dynamics in reward-guided behavior.

blockHMM is implemented at https://github.com/nhat-le/ssm.

**Funding:** This work was supported by US National Institute of Health (NIH) grants R01MH126351 and R01NS130361 (MS), R01MH133066 (MS), R00 EB027706 (MY), Cleveland Clinic and IBM Discovery Accelerator Grant (MY), Army Research Office grant W911NF-21-1-0328 (MS), Paul and Lilah Newton Brain Science Research Award (NML). The funders had no role in study design, data collection and analysis, decision to publish, or preparation of the manuscript.

**Competing interests:** The authors have declared that no competing interests exist.

Here, using a reversal learning task, we showed that mice displayed a much more variable behavior than would be expected from a uniform strategy, suggesting that they mix between multiple behavioral modes in the task. We develop a computational method to dissociate these learning modes from behavioral data, addressing the challenges faced by current analytical methods when agents mix between different strategies. We found that the use of multiple strategies is a key feature of rodent behavior even in the expert stages of learning, and applied our tools to quantify the highly diverse strategies used by individual mice in the task. We further mapped these behavioral modes to two types of underlying algorithms, model-free Q-learning and inference-based behavior. These rich descriptions of underlying latent states form the basis of detecting abnormal patterns of behavior in reward-guided decision-making.

## Introduction

Reversal learning is a behavior paradigm that is often used to study the cognitive processes underlying reward-guided action selection and evaluation of actions based on external feedback [1,2]. Experiments using this task in humans and diverse animal models have contributed to our understanding of the cortical and subcortical circuits that are involved in components of value-guided decision-making such as the evaluation of reward-prediction errors and value [2–6], assessment of uncertainty [7,8], and model-based action selection [9–11]. Detection of aberrant behavioral patterns in reversal learning is critical in clinical diagnostics, as these disruptions are often involved in neuropsychiatric disorders such as obsessive-compulsive disorder, schizophrenia, Parkinson's disease [12–14], as well as neurodevelopmental disorders [15,16].

In uncertain environments, simple models with relatively few parameters can be fitted to predict the behavior of reinforcement learning agents. For example, an often-used approach is to fit the behavior to a reinforcement learning agent with a learning rate parameter, together with an inverse temperature or exploration parameter [4,17–19]. However, recent studies into rodent behavior in reversal learning have revealed more complex behavior in this task, suggesting that simple models might not be sufficient to capture natural behavior. For example, when the level of uncertainty in the environment changes over the course of the experiments, mice can adapt their learning rates according to the statistics of the environment [20,21], suggesting that the learning rate is not fixed across the trial but varies depending on their internal estimates of the environment uncertainty. Furthermore, rodent behavior comprises two concurrent cognitive processes, one reward-seeking component and one perseverative component, which operate on different time scales during the training session [22]. These previous modeling approaches suggest a rich diversity in rodent behavior in the task and the need for sophisticated computational techniques to model the behavior.

A source of behavioral variability that has not been well studied in previous studies is the use of mixed strategies in reversal learning. In other behavioral tasks, state-space modeling has revealed the existence of multiple behavioral states that interchange during sensory discrimination [23,24]. It is unclear whether the use of multiple strategies also exists in rodents' reversal learning behavior, and if so, what components exist in the rodents' behavioral repertoire. It is also unknown from previous studies whether mice use these complex strategies only in difficult reversal learning tasks, or whether complex strategies are also commonly used even in relatively simple, deterministic reversal learning environments.

Here, we investigated these questions using a combination of behavioral experiments and new computational methods to analyze the mixture of strategies in rodent reward-guided behavior. We studied the behavior of mice in a deterministic reversal learning task involving two alternative choices, a simple task that can optimally be solved by a "win-stay, lose-shift" strategy. Despite this simplicity, we found that mice exhibit sub-optimal behavior in the task and deviated significantly from a uniform strategy. To dissociate the components of these mixed strategies, we have built on a previous state-space approach [24] to build a blockwise hidden Markov model (blockHMM) which allows inferring the latent states that govern rodent behavior within single sessions. Using this tool, we classified and characterized different modes of behavior and found that they can be grouped into four main classes: a "low-performance" class, two "intermediate-performance" classes, and a "high-performance" class. Finally, we showed that these diverse modes of behavior can be accounted for by two different models, model-free behavior involving trial-by-trial value adjustments, and inference-based behavior involving Bayesian inference of the underlying hidden world state. These new results and methods highlight the use of mixtures of strategies as a significant source of variability in rodent behavior during reversal learning, even in a deterministic setting with little uncertainty.

## Results

### Mice display sub-optimal behavior in a 100–0 reversal learning task

We trained head-fixed mice on a reversal-learning task involving two alternate actions (Fig 1A). Mice were placed on a vertical rotating wheel [25], and on each trial, they were trained to perform one of two actions, left or right wheel turns. On each trial, one movement was rewarded with probability of 100% and the other with the complementary probability of 0 (Fig 1B). The environments were volatile such that the high- and low-value sides switched after a random number of trials sampled between 15–25 without any external cues, requiring animals to recognize block transitions using only the reward feedback. To ensure stable behavioral performance, we required the average performance of the last 15 trials in each block to be at least 75% before a state transition occurred. We collected behavioral data from $n = 21$ mice that were trained in the task for up to 30 sessions per animal (typical animal behavior shown in Fig 1C).

In the 100–0 environment, the optimal strategy that yields maximum rewards is "win-stay-lose-shift", repeating rewarded actions and switching actions after receiving an error which signals the beginning of the next block. Following this strategy, the accuracy in each block would be 93–96% per block (14/15 to 24/25 depending on the block length). Expert rodent behavior falls below this optimal level (Fig 1D) as their performance asymptotes to only 62% (expert performance range on day 30 across all animals was 30%– 76%), suggesting a source of sub-optimal reversal strategy that fundamentally underlies their behavioral patterns in the task.

Deviations from a perfect win-stay lose-shift strategy could occur due to two reasons: animals might make more errors at the beginning of the block (early, persistent errors), or they might have a sustained error rate even after switching sides (late, regressive errors) [26,27]. To determine the source of the sub-optimality, we examined the number of "initial errors", the average number of trials it took for animals to switch directions per block, and the "late performance", which is their average performance on the last 10 trials of a block. A win-stay lose-shift agent would have 1 initial error and 100% late performance (dashed lines, S1A and S1B Fig). Experimental animals showed both a higher number of initial errors (1.9 ± 0.2 initial errors) and lower late performance (79 ± 4% performance; mean ± standard error for $n = 21$ animals) compared to the ideal win-stay lose-shift agents (S1A and S1B Fig). Overall

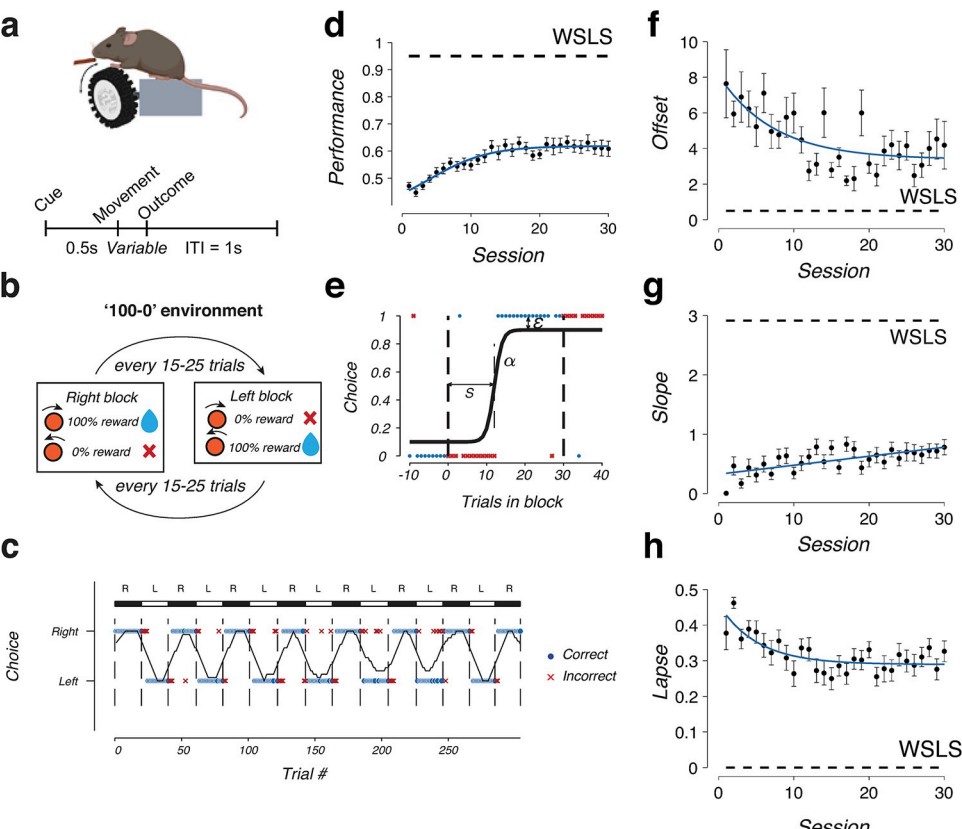

**Fig 1. Mice display noisy behavior in a deterministic reversal learning task.** (a) (Top) Behavioral task setup for head–fixed mice with freely–rotating wheel. Schematic created with biorender.com. (Bottom) Timing structure for each trial, demarcating the cue, movement and outcome epochs. (b) Structure of deterministic reversal learning task in a '100–0' environment. Hidden states alternated between right–states, with high reward probability for right actions, and left–states, with high reward probability for left actions. The block lengths were randomly sampled from a uniform distribution between 15–25 trials. (c) Example behavioral performance of an animal in the reversal learning task, block transitions are demarcated by vertical dashed lines. Dots and crosses represent individual trials (correct or incorrect). Black trace indicates the rolling performance of 15 trials. (d) Session–averaged performance of all mice ($n = 21$) during training of 30 sessions. Dashed line indicates the ideal win–stay–lose–shift (WSLS) strategy. (e) Illustration of the sigmoidal transition function with three parameters: switch offset s, switch slope $\alpha$, and lapse $\varepsilon$. Session–averaged switch offset (f), slope (g), and lapse (h) of all mice (n = 21) during training of 30 sessions. Dashed line indicates the ideal win–stay–lose–shift (WSLS) strategy.

performance was not correlated with side bias (difference in performance between left and right blocks; $R = -0.3$, $p = 0.2$, S1C Fig). Across all mice, there was no significant difference in block performances on left versus right blocks (S1D Fig; $p > 0.05$ for 29/30 sessions, Wilcoxon signed rank test).

To characterize their switching dynamics more precisely, we fitted a logistic regression model with three parameters to observed choices (Fig 1E). These three parameters represent the latent transition between actions: the switch offset $s$, slope $\alpha$, and lapse $\varepsilon$. The switch offset measures the latency of the switch, the slope measures the sharpness of the transition, while the lapse rate signifies the behavioral performance after the transition. A win-stay lose-shift agent would have offset close to 1, very high slope and zero lapse ("WSLS" dashed lines, Fig 1F-H). Expert mice on day 30 instead showed significantly longer switch offset, gentler slope and higher lapse rates ($p < 0.01$, $p < 0.001$, $p < 0.001$ respectively, Wilcoxon signed rank test; Fig 1F–1H). Thus, sub-optimal reversal learning behavior in rodent behavior was due to a combination of both slow switching and high lapse rates.

## Mice display diverse and non-uniform switching dynamics in single sessions

Our logistic regression model assumes rodent behavior is uniform in each session, but it is not clear if this assumption is valid. For instance, if mice use different transition modes within single sessions, the previous analysis would be insufficient to describe the switching dynamics. Supporting the possibility of multiple strategies during the session, we observed highly variable block-by-block performance and lapse rates within behavioral sessions. For example, the block-by-block performance and lapse rates of an animal, E54, were highly variable even at the expert stage (days 26–30 of training; Fig 2A). This variability is much higher than would be expected by a uniform strategy (red error bars; Fig 2A). There was also no apparent of change of strategies between the first and last blocks of the session, suggesting that these sources of randomness occur sporadically during the session and not due to a change in motivation of the animal with satiety states.

This unexpected increase in behavioral variability was consistently observed in our cohort of animals (Fig 2B). For each animal, we computed the "observed" variability in performance, which is standard deviation of performance across all blocks in the final 5 training sessions of the animals (black bars, Fig 2B). We then computed the degree of variability that would be expected from a uniform strategy. To measure this expected variability, we fit a single sigmoidal transition curve to the behavior in these last 5 sessions and generated the behavior of an agent that always executes this transition dynamics in all blocks in these sessions. We again computed the standard deviation in performance of this simulated behavior, and repeating this procedure for $N = 100$ runs yields a distribution of this variability measure of the uniform agent. In 20/21 animals, the observed variability of performance was significantly higher than expected ($p < 0.01$ across bootstrapped distribution), suggesting that rodent behavior is highly non-uniform and that they could be using multiple transition strategies in their behavioral sessions.

## A state-space model to disentangle mixture of transition dynamics in reversal learning

The previous analysis suggests that mice might perform mixtures of transition modes within single training sessions, raising the need to develop more sophisticated analytical techniques to disentangle these behavioral mixtures from behavioral data. We formalized the new analytical framework with a mixture model where individual animals might vacillate between different strategies, switching their choices immediately in some blocks, transitioning more slowly in others, and selecting choices at random toward the end of the session as they became satiated (red, green, and blue shades in Fig 3A, respectively, for a simulated agent). In our framework, each of these strategies might be governed by a separate choice transition function with varying offsets, slopes and lapse rates (sigmoidal curves in Fig 3B). We took advantage of recent developments of state space models that were used to infer discrete latent states from sequences of discrete or continuous variables [24,28,29]. In particular, adapting the previously developed Generalized Linear Model–Hidden Markov Model (GLM-HMM) framework [24] to the reversal learning setting, we assumed that each hidden state determines the parameters of a single sigmoidal transition function (offset $s$, slope $\alpha$ and lapse $\epsilon$), which in turn determines the joint log likelihood of all the choices within each block. We named the approach "block Hidden Markov model (blockHMM)" to indicate the use of hidden states which dictate the evolution of choices throughout the block duration (Fig 3A).

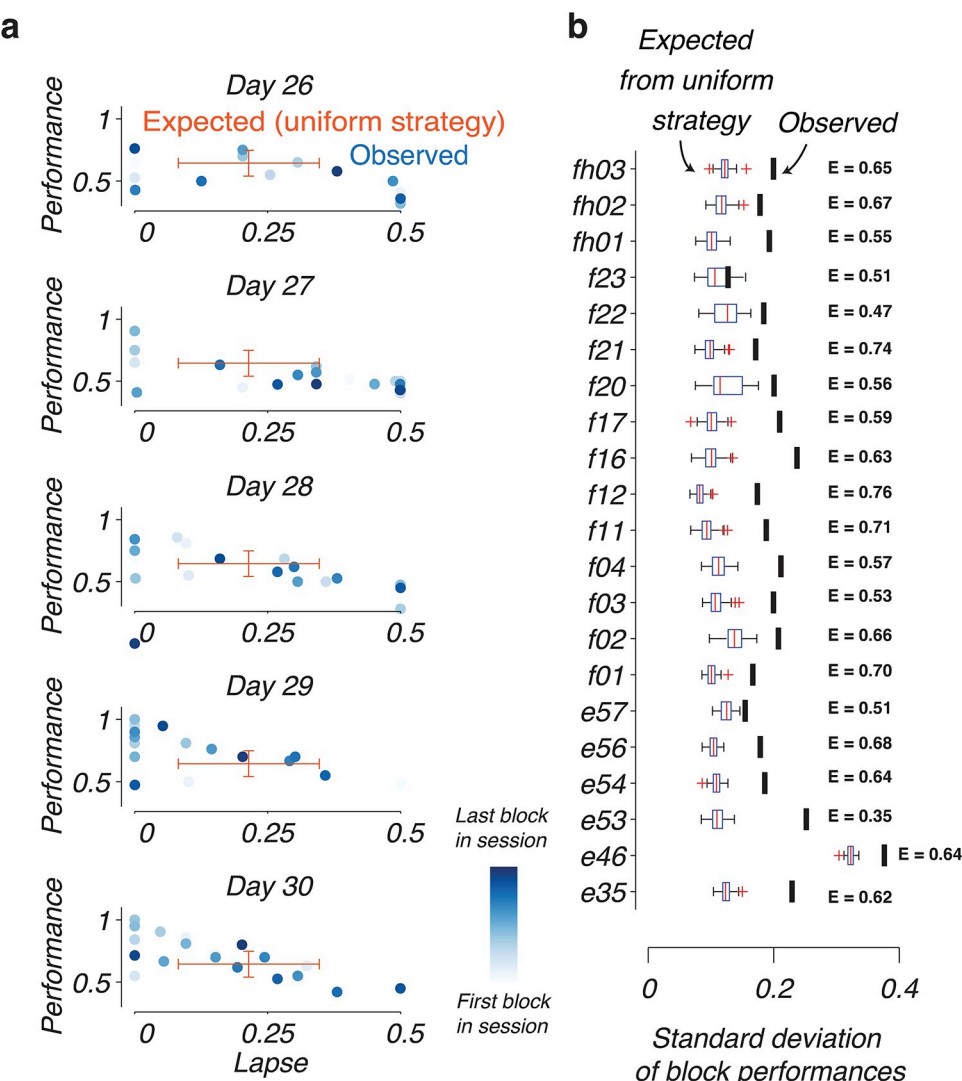

**Fig 2. Non-uniform performance of mice in reversal learning.** **(a)** Example performance of a mouse (E54) in relation to its lapse rate in the last 5 days of training (days 26–30). Individual dots show the combinations of performance and lapse rates in single blocks. Lighter blue dots represent early blocks in the session while dark blue dots are late blocks. Red error bars represent the expected mean and standard deviation in performance and lapse rate assuming the mouse uses a single strategy. **(b)** Comparison of the observed standard deviation of block performances in the final five training sessions (black vertical lines) with the expected standard deviation in performance for an agent that uses a uniform strategy (box plots, $n = 100$ bootstrap runs). Each row represents one of 21 experimental mice (ID of animals shown on the y–axis). The average performance of each animal on the last 5 days of training, $E$, is shown on the right.

More concretely, we assumed that the choice sequences in each block $k$ is governed by an underlying sigmoidal transition function $\sigma_k(t)$, where $t = 0, 1, 2, \ldots$ are the trial numbers within the block (Fig 3A). These transition functions can be parameterized by the switch delay $s_k$, slope $\alpha_k$ and lapse rate $\epsilon_k$ (Eq M.2 and Fig 3B). The discrete latent states $z_i$'s evolve from one block to the next with a Markovian property specified by the transition matrix $P(z_{i+1}|z_i)$ (denoted by arrows in Fig 3A and 3B). The transition function determines the likelihood of all trials within each block (Eq M.3 and Fig 3B). Finally, to fit the model, we used the EM algorithm to maximize the log-likelihood over all observed choices, which is the sum of the log-likelihoods of individual trials (Eq M.4 and Fig 3B).

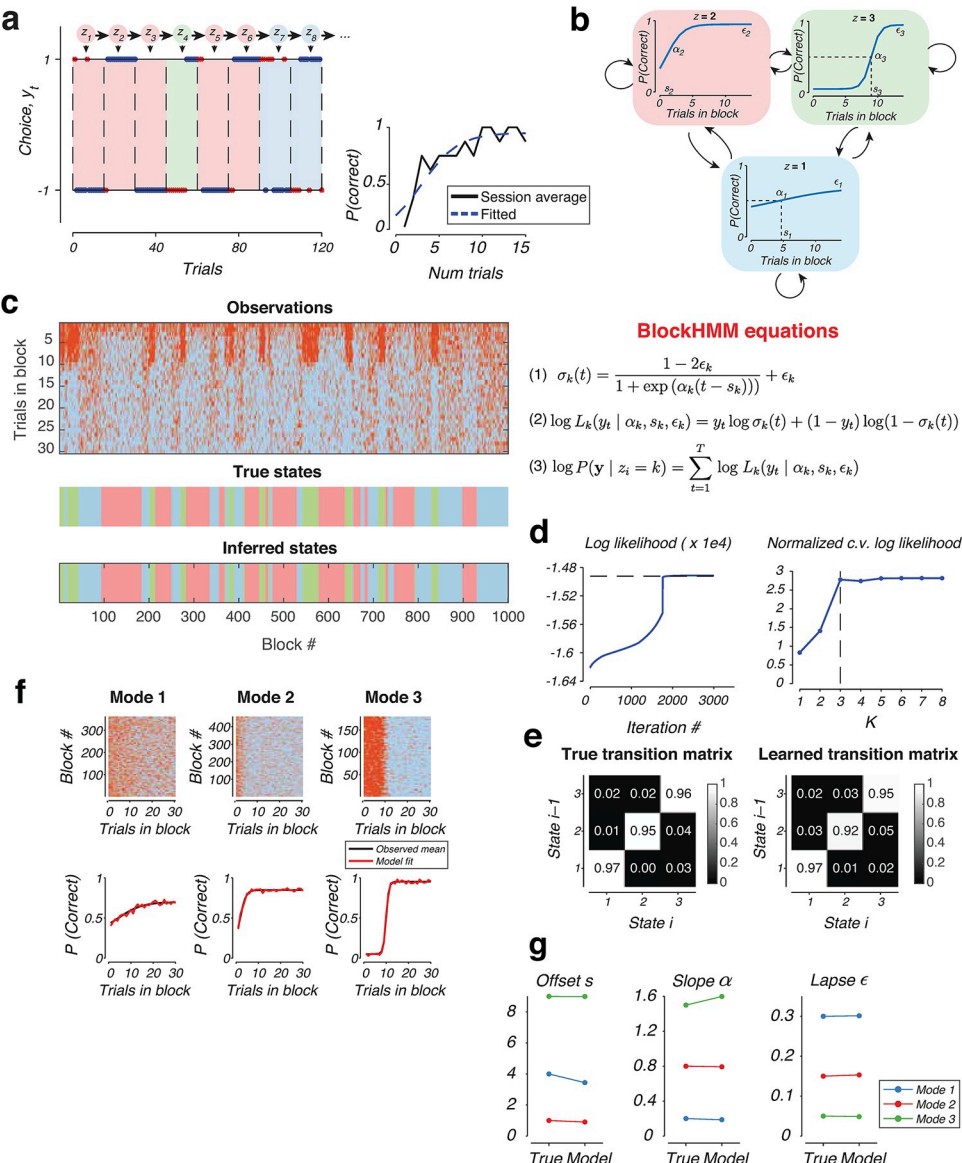

**Fig 3. Block Hidden Markov Model.** a) Behavior of an example agent generated by a Hidden Markov process with K = 3 components. Colored circles represent the underlying hidden states, $z_i$, which evolve according to a Markov chain. Each state (shown by blue, red and green shade) follows a different set of underlying switching dynamics. Blue dots represent correct choices, red crosses represent incorrect choices. (Inset) Average transition function across all blocks of the session (black) together with the fitted sigmoidal curve (blue). b) (Top) Transition functions corresponding to each of the three hidden states, $z_i = 1, 2, 3$. Each sigmoidal curve can be parameterized by three features, the slope $\alpha_i$, offset $s_i$, and lapse $\epsilon_i$. Arrows represent transition probabilities between the states. (Bottom) Eqs of the blockHMM generative model. Each hidden state governs the choice sequence in each block according to the sigmoidal transitions (Eqs 1 and 2). The log–likelihood of the observed choices in the block is the sum of the log–likelihoods of individual trials (Eq 3). c) (Top) Example behavior in 1000 blocks of trials generated by the same blockHMM mixture shown in panels a and b. Each column represents one block, with trials 1 to 30 of each block running from top to bottom. Red represents incorrect choices and blue represents correct choices. (Middle) True states that underlie the behavior shown in the top panel. (Bottom) Inferred latent states by the blockHMM fitting procedure. d) (Left) Convergence of the log–likelihood during model fitting in panel c to the true log–likelihood of the data (dashed line). (Right) Dependence of cross–validated log–likelihood on the number of components, K. e) True and inferred transition matrices for the behavior shown in panel c. f) Grouping of blocks of trials according to the inferred state after the model fitting with K = 3 HMM components. (Top) Raw behavioral performance grouped by the identity of the inferred latent state. (Bottom) Average transition function and fitted sigmoidal curve for all blocks that share the same inferred state. g) Comparison of true and inferred parameters for the three components of the behavior shown in panel c.

Our synthetic agent (Fig 3A) was simulated according to a blockHMM process with $K = 3$ hidden states. State $z = 1$ (blue) corresponded to a random mode of behavior with a flat transition function, $z = 2$ (red) corresponded to a sigmoidal curve with a fast offset, and $z = 3$ (green) involved a sharp but delayed switching of actions. We generated the behavior of this agent over 1000 blocks (Fig 3C), and fitted the blockHMM model to the observed choice sequences of the agent. The log-likelihood of the fit converged to the true log likelihood value (Fig 3D, left). To determine the best number of latent states for the model, we trained the model on 80% of the blocks and evaluated the log-likelihood on the remaining 20% of the blocks. Inspecting the normalized cross-validated log-likelihood, we found that the optimal number of clusters was $K = 3$, agreeing with the ground-truth value (Fig 3D, right). At the end of the fitting procedure, blockHMM recovered the correct transition matrix (Fig 3E), as well as the parameters of the transition function in each mode (Fig 3F–3G). Importantly, the inferred latent states closely matched the true states that underlie the behavior (Fig 3C, bottom panels).

## Components of mixtures of strategies of expert mice

We applied the algorithm to analyze the behavior of expert mice ($n = 21$) in the task. The model was fitted to individual animal behavior and maximum cross-validated log-likelihood was used to determine the number of components, $K$, that maximized this metric (S2A Fig). The number of behavioral modes ranged from 2 to 6 in our experimental cohort, and did not differ significantly between males and females (S2B Fig). Transition matrices for individual mice are shown in S3 Fig.

The shapes of the blockHMM transition curves in experimental animals were highly diverse and showed various types of transition dynamics (Fig 4A). Some modes had slow slopes, others had fast switching, and others had high lapse rates. The distribution of mode performance was multi-modal and could be clustered into three groups (Fig 4B), with low performance (below 65%), intermediate performance (65–84%) and high performance (above 84%). These performance levels correspond to three distinct regimes of behavior with different transition dynamics (Fig 4C and 4D). "Low performance" modes are random behavior with flat transition curves with high switch offsets or high lapse rates indicating random behavior with no switching (blue; Fig 4C). "Intermediate performance" modes can be further broken down to two subgroups: one with high switch offset (magenta; Fig 4C) and the other with high lapse rate (yellow; Fig 4C). "High performance" modes have both low offsets and low lapse rates (green; Fig 4C). We verified that each animal executes a mixture of behavior from multiple regimes instead of behavior in a single domain (Fig 4E). Indeed, 17/21 mice (81.0%) used a combination of low, medium and high-performance behavioral modes. The remaining 4/21 mice (19.0%) used low and high-performing behavioral modes with no intermediate modes. This shows that rodents tend to mix between diverse types of strategies during reversal learning. There was no apparent difference in composition of strategies between males and females (S2C Fig). Similarly, no significant sex differences were found in the overall performance in the task (S1E and S1F Fig, $p = 0.97$ for sessions 1–10, $p = 0.7$ for sessions 11–20, $p = 0.2$ for sessions 21–30, Mann-Whitney U-test).

## The diversity of transition modes is accounted for by the spectrum of model-free and inference-based strategies

Our analytical approach decomposes rodent behavior in the reversal learning task into a number of switching modes, each characterized by a respective transition curve. Some of these transition modes have very fast offsets and close to zero lapse rates, resembling the behavior of a win-stay lose-shift agent. However, other modes were sub-optimal and cannot be explained

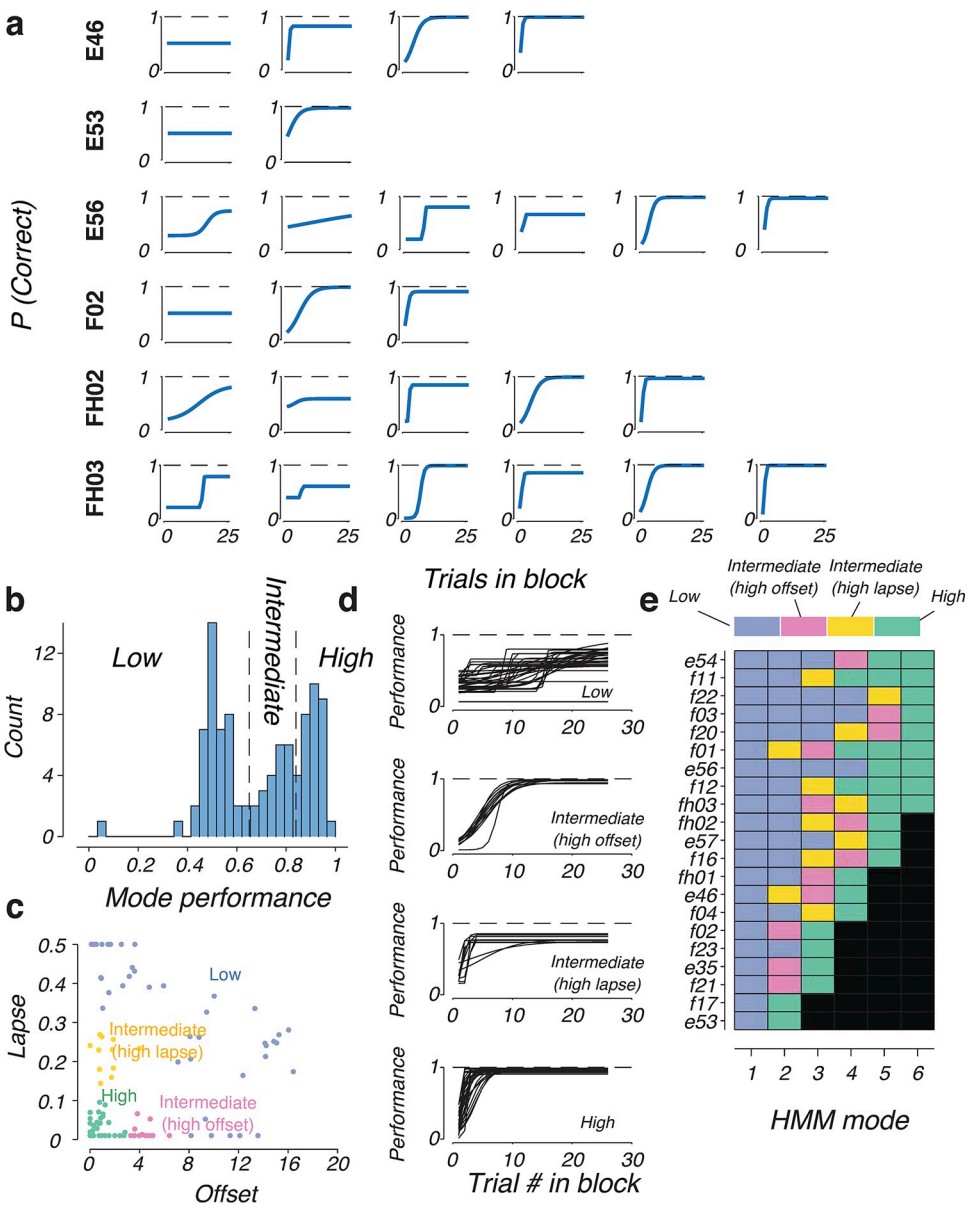

**Fig 4. Component mixtures of expert mice. (a)** blockHMM transition curves of six representative mice (E46, E53, E56, F02, FH02, FH03). The number of behavioral modes ranged from two to six in all our experimental mice. In each row, transition curves are sorted in increasing order of performance. **(b)** Distribution of block performances across behavioral modes. Behavioral modes were divided into three groups based on this distribution: low performance (<65%), intermediate (65–84%), and high (>84%). **(c)** Distribution of two transition parameters, offset and lapse of the low–performance, medium–performance and high–performance groups. The low performance regime (blue) had a high offset and lapse rate. The intermediate performance regime consists of two sub–groups: one group (yellow) had a high lapse rate but a low offset, and the other (pink) had a high offset but a low lapse rate. The high–performance regime (green) had a low lapse rate as well as an offset. **(d)** Block transition dynamics of the four behavioral regimes identified in c. **(e)** Types of behavioral modes found in the behavior of each experimental mouse (n = 21). All animals performed a mix of behavioral modes ranging from low to high performance.

by such a strategy. What types of underlying algorithm might give rise to these modes? We will show next that the diversity in transition modes can be sufficiently accounted for by the space of Q-learning and inference-based behavior, two types of algorithms that are frequently discussed in the literature of reversal learning.

More concretely, the spectrum of transition functions might be accounted for by the variability between two classes of agents, Q-learning and inference-based agents. Q-learning is a model-free learning strategy that performs iterative value updates based on external feedback from the environment (Fig 5A, top). In the reversal learning task, the agents maintain two values for left and right actions, $q_L$ and $q_R$. The value of the chosen action on each trial is updated according to

$$q_i \leftarrow q_i + \gamma(r - q_i) \tag{1}$$

where $r$ is the trial outcome (0 for errors or 1 for rewards), and $\gamma$ is the learning rate parameter. We additionally assumed that the agent adopts an $\varepsilon$-greedy policy, choosing the higher-valued action with probability 1 - $\varepsilon$, and choosing actions at random (with probability 50%) on a small fraction $\varepsilon$ of trials.

In contrast, "inference-based" agents select actions by inferring the world's hidden state, i.e., which side is more rewarding, on each trial (Fig 5A, bottom). The internal model of these agents consists of two hidden states, $L$ and $R$, that determine whether the "left" or "right" action is associated with higher reward probability. The transitions of these hidden states are approximated by a Markov process with probability $P_{switch}$ of switching states and $1 - P_{switch}$ for remaining in the same state on each trial. Given this model and observed outcomes on each trial, the ideal observer can perform Bayesian updates to keep track of the posterior distribution of the two states (see update equations in *Methods*). The agent then uses the posterior over the world states to select the action that maximizes the expected reward on that trial.

To understand the correspondence between the type of algorithm (Q-learning or inference-based) and the shape of the transition function, we will break down our analysis into two steps. We first built a forward model by performing a simulation of behavior that is exhibited by Q-learning and inference-based agents with different model parameters. This analysis characterizes and quantifies the features of the transition dynamics shown by each agent. Then, we evaluated whether it is possible to infer the underlying strategy based on the observed transition function using a decoding approach.

## Forward simulations

For forward simulations, the behavior of model-free agents was simulated for a range of parameters where $0.01 \le \gamma \le 1.4$, and $0.01 \le \epsilon \le 0.5$, and inference-based agents were simulated for a parameter range $0.01 \le P_{switch} \le 0.45$ and $0.55 \le P_{rew} \le 0.99$ (example simulations shown in Fig 5B). As expected from the roles of the parameters from previous literature, transition curves reflect the variations of these parameters along principled axes. For instance, for model-free agents, increasing $\gamma$ leads to faster switch offsets (S4A–S4C Fig), while varying $\epsilon$ predominantly affects the lapse rates of the sigmoidal transitions (S4B and S4D Fig). Inference-based agents are clearly distinguished from model-free behavior by their small lapse rates (S5 Fig). Their behavior varies along an axis that corresponds to the volatility of the environment. As $P_{switch}$ and $P_{rew}$ increase, the internal model assumed by the agents become increasingly volatile. This makes agents more sensitive to errors and hence resulting in faster switch offsets (S5B Fig). Despite these variations in the latency of the switch, the lapse rates of inference-based agents generally remain close to zero.

## Backward inference

Although there is a wide variation in the features of the transition curves that are exhibited by the various agents, the four types of transition modes that we previously observed in rodent behavior can be seen in particular regimes of either Q-learning or inference-based behavior.

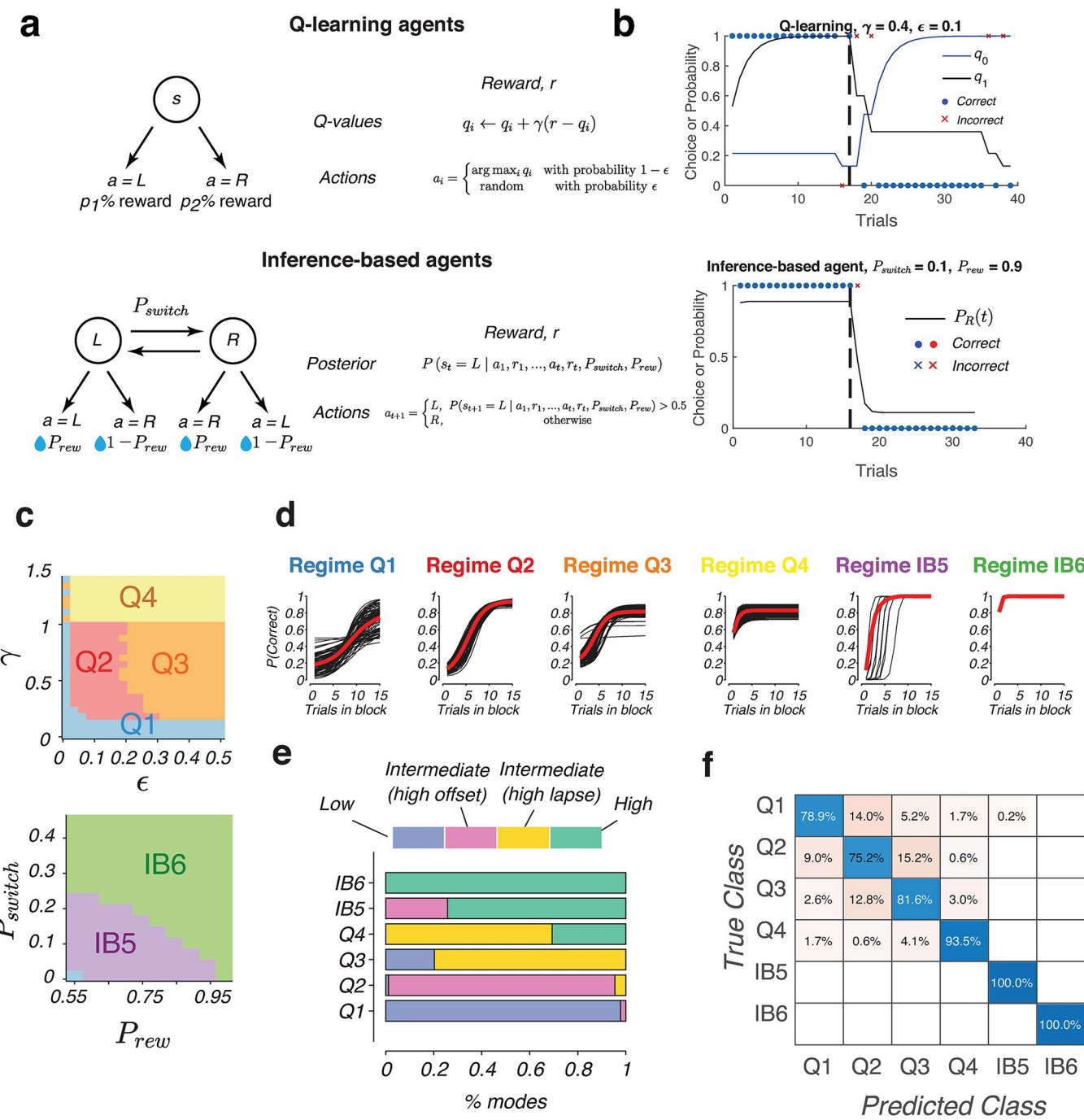

**Fig 5. Mapping transition dynamics to underlying behavioral strategies.** (a) Implementation of Q–learning (top) and inference–based algorithms (bottom) for simulating choice sequences of artificial agents. (b) Example behavior of simulated Q–learning (top) and inference–based agents (bottom). Each dot or cross represents the outcome of a single trial. In the Q–learning plot, black and blue traces represent the values of each of the two actions. In the inference–based plot, black trace represents the posterior probability of the right state $P(s_t = R \mid c_1, r_1, \ldots, c_{t-1}, r_{t-1})$. (c) We performed a computational simulation of an ensemble of Q–learning and inference–based agents taken from grids that spanned the Q–learning parameter space (top), or the inference–based parameter space (bottom). Based on the results of the simulations, the spaces were clustered into six groups (represented by different colors), that showed qualitatively different behavior. (d) Transition functions grouped according to the behavioral regime Q1–4, IB5–6. Black lines represent single agents and red trace represents the mean across all the transition functions in each group. (e) Behavioral regime composition of each of the six algorithmic domains (Q1–4, IB5–6). (f) Cross–validated confusion matrix showing the classification performance of a k–nearest neighbor (kNN) classifier trained to predict the class identity (Q1–4, IB5–6) based on the observed transition curve. Diagonal entries show the accuracy for each respective class.

(1) "Low-performance", random behavior is seen in the model-free agents with low learning rate. (2) "Intermediate-performance" behavior with high offset is seen in model-free agents with low learning rate and low exploration rate. It is also seen in inference-based agents in the stable regime. (3) "Intermediate-performance" behavior with high lapse is seen in model-free agents with high exploration rates and high learning rates. (4) "High-performance" behavior is seen in the inference-based agents in the volatile regime.

We aim to establish a more precise mapping between the algorithmic spaces (Q-learning and inference-based) and transition dynamics exhibited by the agents. To simplify this mapping, we clustered the sigmoidal transition features of all simulated inference-based and Q-learning agents into groups that display similar transitions. Each agent's transition curve was reduced to four features–the fitted switch offset, slope, lapse, and overall performance which were used to perform an unsupervised clustering into six groups (S6 Fig). These groups clustered together in the parameter spaces (Fig 5C). The transition curves of these six classes resembled the types of transitions that were observed in rodents' HMM modes (Fig 5D). Matching these groups to the corresponding regime on the parameter spaces, we found that "low-performance" behavior was seen primarily in the Q1 regime (Fig 5E) which had the lowest learning rate $\gamma$. "Intermediate-performance", high offset was seen in the Q2 regime which had intermediate learning rate but low exploration $\varepsilon$. "Intermediate-performance" with high lapse was seen in regime Q3, with the same learning rate as Q2 but higher exploration. It is also seen in regime Q4, the class which contains various agents with very high learning rate ($\gamma > 1$). "High-performance" behavior was seen in the inference-based agents (IB5-6). The regime IB5 also contains a small number of agents with intermediate-performance and high offset.

To validate the utility of these classes, we trained a *k*-nearest neighbor (kNN) classifier to predict the class identity (Q1-4 or IB5-6) of synthetic agents given the observed transition curve. We found that the classifier performed with a high accuracy of 88% on a held-out test set (Fig 5F), compared to a chance performance of 17%. Altogether, these results establish a consistent and reliable mapping from the underlying strategies in the model-free or inference-based parameter spaces, to the observed transition functions that can account for the diversity of rodent behavior during reversal learning.

## Changes in the composition of strategies during task learning

We next investigated changes in the composition of strategies during the learning of the task. Behavioral modes of animals were first re-classified into the six behavioral strategies, Q1-4, IB5-6 (Fig 6A and 6B). Animals typically used a combination of Q-learning and inference-based modes throughout their training sessions. However, there was some variability in the types of strategies used by different animals. A small subset of animals, such as f16 (Fig 6C), consistently employ model-free strategies during execution of the task. Other animals, such as f11 (Fig 6D) started with a model-free strategy but transitioned to inference-based modes with learning. Remarkably, even in the expert stage (day 30 of training), the animal never operated fully in the inference-based regime but continues to execute a mixture of strategies. This was a common feature of many animals that managed to reach the inference-based stage (such as animal e46, e54, e56, f01, f11, f12, fh02, fh03, Fig 6A. The compositions of blockHMM decoded strategies of individual animals are shown in S7 Fig, together with the individual session performances).

This shift from model-free to inference-based behavior was seen in the average mixture composition of the animals (Fig 6E). On average, animals started training with a significant fraction of the Q1 mode and smaller fraction of Q4 (56% in Q1 and 24% in Q4, averaged across

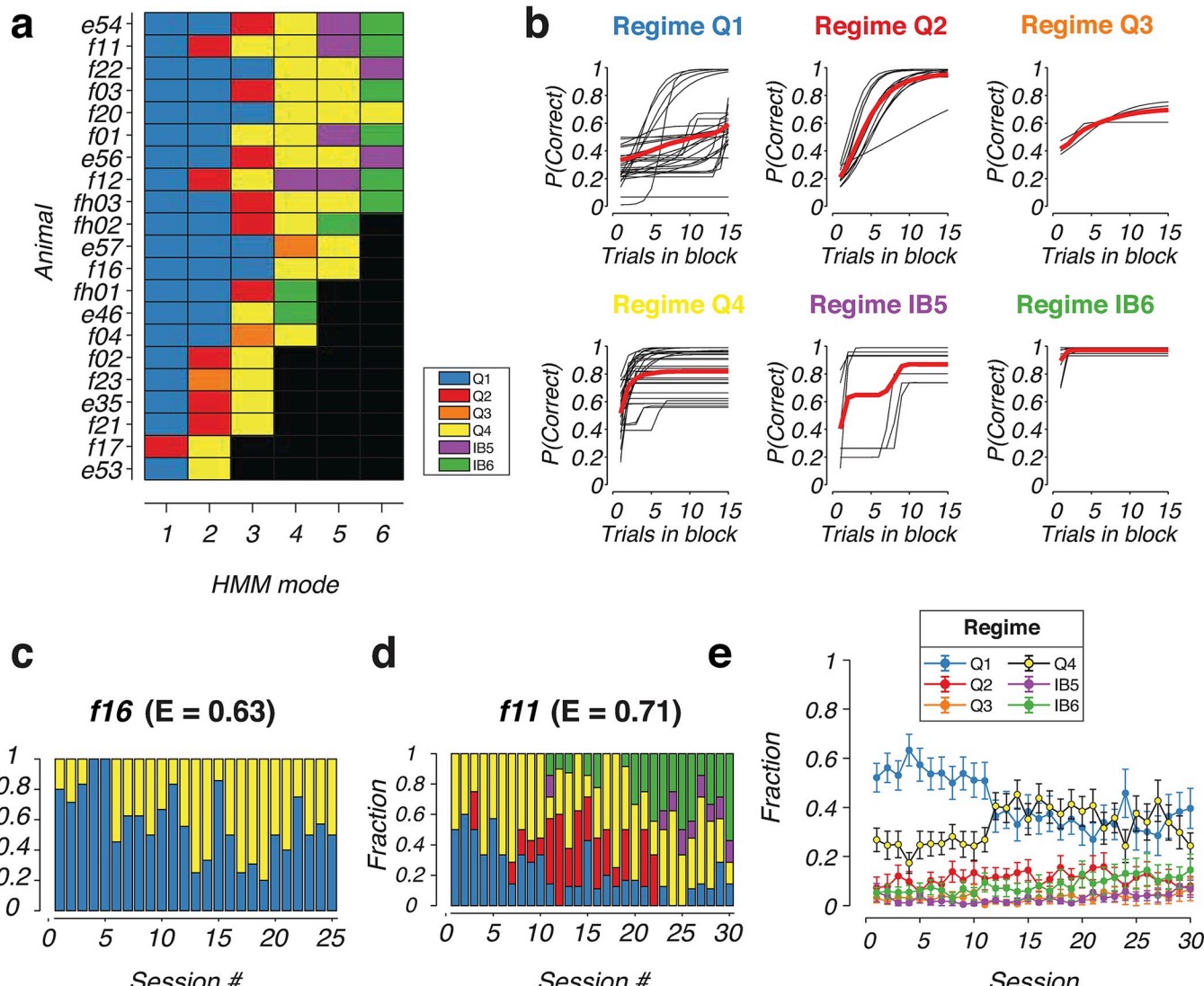

**Fig 6. Mice use combination of model–free and inference–based strategies in reversal learning.** (a) Composition of blockHMM mixtures for individual animals. Each row represents one mouse with ID shown on the left. The color of each square represents the decoded behavioral regime of each HMM mode (Q1–4, IB5–6). The number of blocks for each animal, K, was selected by cross–validation and are sorted here in descending order. (b) Transition function of HMM modes for all animals, grouped according to the decoded behavioral regime. (c, d) Distribution of HMM modes for two example animals, f16, and f11, which displayed vastly different behavioral strategies with learning. The average performances of the two animals on the last 5 days of training, *E*, are shown. (e) Average frequency of HMM modes for n = 21 experimental animals (mean ± standard error) showing the average evolution of behavioral mixtures over the course of training.

days 1–5). Over the course of training, the mixture of behavioral strategies slowly shifted from Q1 to Q4, such that around day 15, there is a higher fraction of Q4 than Q1 mode (39% in Q4 compared to 35% in Q1, averaged across days 16–20). This shift reflects an average increase in learning rate in the Q-learning regime. At the same time, the fraction of inference-based modes, IB5 and IB6, was low at the beginning (3% in IB5 and 6% in IB6 averaged across days 1–5), but continuously increased as animals gained experience with the task. At the expert stage, there was an increase in fraction of blocks in the inference-based mode (7% in IB5 and 15% in IB6 on day 30), but the mixture of strategies still remained with Q1 and Q4 being the primary Q-learning modes of the animals. These patterns of strategy mixture were consistent

between male and female mice, with no statistically significant difference in composition between sexes (S8 Fig; $p > 0.05$ for all modes and session groups, Mann-Whitney U-test). Overall, these ubiquitous use of mixtures of strategies, which were distinctive both in naïve and expert animals, further underscore the importance of our approach to dissociate and characterize the features that constitute individual modes of behavior.

### Identification of early switches in blockHMM modes

Thus far, our analysis has assumed that animals execute transitions once the block reversals have happened, with the switch offset $s$ constrained to be positive. By relaxing this assumption, we investigated whether mice can anticipate upcoming block switches after prolonged task exposure, consistent with previous findings [8,30]. In each block, we included three additional trials prior to the block transitions, and repeated the blockHMM fitting procedure. The distribution of the fitted parameters of the modes was similar to the original results (S9A–S9C Fig; compared to Fig 4B–4D). Notably, a subset of high-performing modes had offset $s < 0$, suggesting behavioral transitions prior to the actual block reversal (S9B Fig). These "high-performance, early" switches ($s < 0$) displayed distinct transition dynamics compared to the "high-performance, regular" switching modes ($s \geq 0$; S9C, bottom panel). Together, these early switch modes comprise a low but constant proportion of all blocks in the animals' switching behavior (S9D Fig).

### Discussion

Using a simple version of the reversal learning task which alternates between 0% and 100% reward probabilities, we demonstrated several sources of non-ideal behavior in mice in this task. Rodent behavior deviates from the performance of a win-stay lose-shift agent in several ways. The average transition of mice was both slower at the beginning of the block (higher perseverative errors), and their error rates remained high even after switching (higher regressive errors). This behavior was better than a random strategy, but worse than an optimized agent, consistent with recent results [20,22]. The non-zero regressive errors in the deterministic environment of our task shows that mouse behavior is more exploratory than would be expected for a matching behavior [31–33].

Another source of unexpected behavior in mice was the use of multiple strategies. We showed evidence that this was a ubiquitous feature of mouse behavior, as 20/21 of our animals showed much higher variability in performance than would be expected by a uniform strategy. Previous studies have investigated the use of different learning rates [20], or different components (reward-seeking and perseverative) [22] of behavior in rodents in the context of reward-guided behavior. Our finding that non-uniformity of behavior is seen even in the context of the 100–0 reversal learning task suggests the widespread use of multiple strategies, even in simple tasks and in deterministic environments. Thus, mixed strategies can be seen in simple behavior that need not involve drastic changes in the statistics of the environment [24].

Through the modeling of different modes of behavior that constitute these mixtures of strategies, we introduce a more general framework than previous studies to disentangle the components of these mixtures. In addition to allowing the learning rates to vary between blocks, we also considered variations in the lapse rates of the transition, an important parameter in rodent behavior [34–38]. We developed a novel state-space model, blockHMM, to segment of behavior during the session into blocks of trials that are governed by different underlying states. Our model builds on recent modeling approaches which are used to infer discrete latent states that underlie neural dynamics [28], natural behavior [29], and behavior in decision-making tasks [24,39].

Applied to rodent behavior, blockHMM reveals diverse type of dynamics exhibited by rodents in the different transition modes. We found four distinct types of transitions in the rodent behavior repertoires: a "high performance" mode with fast switching and low lapse rate, similar to win-stay lose-shift behavior, a "low performance" mode where performance within the block was close to random, and two "intermediate-performance" modes, one with high offset and the other with high lapse rates. Rather than being dominated by any single type, typical mouse behavior involves a combination of these different behavioral modes (Fig 4B–4D). Thus, blockHMM reveals a dynamic composition of rodent strategies and provides descriptive measurements of each transition mode being used. By considering the possibility of negative offsets ($s < 0$), we uncovered a subset of modes where switches occur before the actual block reversal. These modes could arise as mice anticipated upcoming switches [30,40], or due to exploration of the unrewarded actions, a common feature of mouse behavior [34,41]. BlockHMM provides an important tool to identify these blocks for the investigation of neural mechanisms of early switching.

The diverse transition modes were then matched to the behavior of two types of algorithms, Q-learning and inference-based behavior, two models that are frequently used for analysis of choice sequences in reversal learning tasks. Q-learning is a type of reinforcement learning model that has been often used for probing representation of action values in numerous brain regions [4,20,31,42–46], while inference-based behavior has been investigated in studies of internal models in the brain during tasks involving hidden states [8,47]. The transition behavior of simulated Q-learning and inference-based agents indeed resembled the four types of behavior that mice exhibit in their transition modes. Using computational simulations, we showed that Q-learning and inference-based behavior can be grouped into six regimes, Q1-4 for Q-learning and IB5-6 for inference-based. This classification helps establish a mapping between behavioral algorithm and observed transition mode: low-performance modes primarily correspond to Q1, intermediate-performance, high offset corresponds to Q2, intermediate-performance, high lapse corresponds to Q3 and Q4, while high-performance mode corresponds to the inference-based regimes (IB5-6) (Fig 5E). As a transitional regime from Q-learning to inference-based learning, Q4 consists of both intermediate- and high-performance modes. Decoding performance onto these four regimes were high (87% compared to a chance performance of 17%; Fig 5F), highlighting the robustness of this classification scheme.

Consistent with previous findings, rodent behavior shifted from model-free learning modes to inference-based behavior with training [7,10,48] (Fig 6E). However, a notable additional finding from our study is that although the frequency of inference-based modes did increase, inference-based behavior never became dominant even at the expert stage of training. Instead, both model-free and inference-based modes continued to co-exist at this stage. This suggests that mixing between model-free and inference-based behavior could be a key feature of rodent behavior that is preserved despite familiarity with the task. This mixture of strategy was seen for both male and female mice, with comparable number of modes and composition between the two sexes (S2B, S2C and S8 Figs). Although differences in these metrics were not statistically significant, blockHMM provides a rich set of descriptive parameters characterizing the transition dynamics and state transition probabilities that can be a powerful tool for future studies which aim to understand sex differences in behavioral strategies in reversal learning [49–52].

An interesting question this raises is why this use of mixtures of strategies might be adaptive for rodents in their natural environments. One hypothesis is that animals might fail to sustain the memory of action values or internal models of the environment structure over time [18,53]. For instance, the memory of the unchosen action values, or of the internal model parameters ($P_{switch}$ and $P_{reward}$) might degrade over time. This requires exploratory modes of

behavior (Q-learning) to maintain and reconstruct these values. Another hypothesis is that mice continuously revert back to model-free behavior to verify and re-build their internal models of the task environment. This type of transient switches to the exploratory regime would be adaptive in circumstances where the environments frequently change between different types of models such that persisting with a fixed model might decrease the chance of survival. For example, this might be seen in situations where the presence of predators might be unpredictable with patterns shifting between seasons, or where when food sources might vary drastically over time due to unpredictable weather patterns.

Overall, our study highlights the use of multiple transition modes in reversal learning and calls for a focus on the problem of state segmentation in rodent behavioral studies. The methods developed in the paper can be leveraged in investigations of the neural mechanisms that govern these distinct modes, as well as the plasticity of these circuits during the transition between model-free and inference-based behavior. These two types of behavior involve vastly different neural circuits that engage different sub-divisions of the basal ganglia and frontal cortex [5,7,54,55]. Our modeling framework here highlights the highly dynamic nature of the strategy switches, suggesting that switches between these sub-circuits in the brain might occur frequently during single behavioral sessions. Since encoding of task variables is highly state-dependent in many behavioral contexts [56–60], the state information provided by the blockHMM might be particularly valuable for the investigation of state-dependent encoding of task variables such as choice, reward, and value varies across different inferred hidden states. More broadly, blockHMM can also be used to analyze the behavior and neural activity involved in probabilistic foraging tasks [7,44], as well as decision-making tasks that involve alternating blocks with opposing rule sets [61–63]. The state segmentation approach will also be a valuable tool for perturbation experiments, with the power to reveal shifts in composition, order or transition probabilities between these modes, thus augmenting existing methods for a much richer and complete view of rodent behavior during reversal learning.

## Materials and methods

### Ethics statement

All experimental procedures performed on mice were approved by the Massachusetts Institute of Technology Animal Care and Use Committee and conformed to National Institutes of Health guidelines.

### Animals

Mice were housed on a 12 h light/dark cycle with temperature (70 ± 2 ˚F) and humidity (30–70%) control. Animals were group-housed before surgery and singly housed afterwards. Adult mice (2–6 months) of either sex were used for these studies (12 females and 9 males). Mice belonged to four lines: C57BL/6J: 7 animals, Ai184D (B6.Cg-Igs7tm148.1(tetO-GCaMP6f, CAG-tTA2)Hze/J), Jackson #030328: 5 animals; Ai162D (B6.Cg-Igs7tm162.1(tetO-GCaMP6s, CAG-tTA2)Hze/J), Jackson #031562: 7 animals; B6.129(Cg)-Slc6a4tm1(cre)Xz/J, Jackson #014554: 2 animals.

### Surgical procedures

Surgeries were performed under isoflurane anesthesia (3–4% induction, 1–2.5% maintenance). Animals were given analgesia (slow release buprenex 0.1 mg/kg and Meloxicam 0.1 mg/kg) before surgery and their recovery was monitored daily for 72 h. Once anesthetized, animals were fixed in a stereotaxic frame. The scalp was sterilized with betadine and ethanol. The skull

was attached to a stainless-steel custom-designed headplate (eMachines.com) using Metabond. Animals were allowed to recover for at least 5 days before commencing water restriction for behavioral experiments.

## Behavioral apparatus and task training

The training apparatus and software for running the experiments were adapted from the Rigbox framework for psychophysics experiments in rodents [64,65]. Mice were head-fixed on the platform (built from Thorlabs hardware parts) and their body placed in a polypropylene tube to limit the amount of movement and increase comfort. Their paws rested on a vertical Lego wheel (radius 31 mm) which was coupled to a rotary encoder (E6B2-CWZ6C, Omron), which provided input to a data acquisition board (BNC-2110, National Instruments). The data acquisition board also provided outputs to a solenoid valve (#003-0137-900, Parker) which controlled the water reward delivery.

After mice recovered from surgery, they were placed under water restriction for 1 week, with daily water given by HydroGel (Clear $H_2O$). The initial amount of HydroGel was equivalent to 2mL of water a day, and this decreased gradually until mice received an amount equivalent to 40 mL/kg each day. Mice were weighed weekly and monitored signs of distress during the course of training. In the case of substantial weight loss (>10% loss weekly) or decrease in body condition score, the restricted water amount was increased accordingly. Mice were handled daily during the initial 1-week water restriction period for ~10 minutes each day. They were then allowed to explore the apparatus and given water manually by a syringe on the platform. If mice did not receive their daily water amounts during training, they were given the remaining amount as hydrogel (Clear $H_2O$) in their home cage.

When mice were comfortable with the setup, they were head-fixed on the platform and given small water rewards of 4 μL from a lick spout every 10 seconds, for a total duration of 10 minutes. This duration was increased to 20 minutes, and 40 minutes on the two subsequent days. The wheel was fixed during this protocol. On the next day, mice were trained on the *movementWorld* protocol, with the wheel freely moving. Here, each trial was signal with an auditory tone (0.5s, 5 kHz), following which movements in any direction crossing the movement threshold of 8.1˚ rotation were rewarded with 4 μL of water. Mice then had to remain stationary for 0.5 s before the next trial starts. This discouraged a strategy of continuous rotation of the wheel.

After mice became comfortable with this stage and consistently obtained at least 0.6 mL of water each session, they were taken to the final task stage, *blockWorldRolling*. We started tracking the performance of animals when they entered this stage, with session 1 being the first day of training on *blockWorldRolling*. Animals were trained until no performance improvements were observed, for a maximum of 30 sessions.

In the *blockWorldRolling* stage, each trial began with an auditory tone (0.5s, 5 kHz). During a delay period of 0.5 s from the trial tone onset, movements of the wheel were discounted. After this window, the movement period started, where movements of the wheel past a specified threshold were recorded. The threshold was fixed at 8.1˚ in the first session of *blockWorldRolling* and subsequently increased to 9.5˚, and 10.8˚ on the next days. The trials were grouped into blocks of trials of 15–25 trials, with lengths of the blocks sampled uniformly at random. The blocks alternated between the "left" and "right" state. In the "left" state, left wheel turns were rewarded with probability 100% and right wheel turns were not rewarded. In the "right" state, right wheel turns were rewarded with probability 100% and left wheel turns were not rewarded. If mice made the correct movement, they were given a 4 μL water reward. For unrewarded trials, a white noise sound was played for 0.5 s, followed by a time-out of 1 s. After

the trial feedback was given, an inter-trial interval (ITI) of 0.5 s elapsed before the next trial started. The ITI was gradually increased to 1 s once animals performed well in the task. If mice didn't make a choice within 20 seconds, the trial was aborted, signaled by a white noise and 1-s time-out period (similar to an error trial). After the length of the block has passed, if the rolling performance of the animal in the last 15 trials was above 75%, the state of the block would flip and the next block continued. Otherwise, the block continued until the rolling performance in the last 15 trials in the block passed 75%. The session is terminated when the average reaction time in the final 5 trials of the session went above 10 seconds.

## Analysis of animal performance

Performance in each session was defined as the number of rewarded trials divided by the total number of trials in the session. Within each block, the number of persistent errors was defined as the number of errors at the start of each block before a correct response was made in that block. The late performance was defined as the fraction of correct responses in the last 10 trials of the block. The left–right bias in performance of each animal (S1C and S1D Fig) was defined as the fraction of correct responses in all trials on leftward blocks minus the fraction of correct responses in all trials on rightward blocks.

To parametrically describe the rate of switching of animals, we calculated the average accuracy $a_n$, where $n = 1, 2, 3, \ldots$, defined as the average accuracy of all trials occurring at position $n$ in the block, across all blocks of the session. We then fitted a logistic regression model of the form

$$a_n = \epsilon + \frac{1 - 2\epsilon}{1 + exp(-\alpha(n - s))} \tag{M.1}$$

where $\epsilon$, $\alpha$ and $s$ are free parameters representing the lapse rate, slope and offset, respectively. The parameters were jointly fit with the Python function scipy.optimize.minimize(), with constraints $s \geq 0$, $\alpha \geq 0$, $0 \leq \epsilon \leq 0.5$.

To compare the animals' performance and parameters to a benchmark, we simulated the behavior of a "win-stay, lose-shift" agent which makes a single mistake at the beginning of each block. This agent was simulated for a total of 25 blocks. We calculated the number of persistent errors and late performance the same way as for mice. Similarly, the logistic regression model was fitted to the win-stay lose-shift agent's performance in the same way as described above.

## Quantification of performance variation

To quantify the variability in block-wise performance for each animal, we computed the block-by-block performances by determining the fraction of correct choices in each block. We then computed the standard deviation of these block performances across all blocks in the last 5 training sessions of each animal. This gives us the "observed" standard deviation $s_{observed}$ for each animal (black vertical lines in Fig 2B).

We then compared $s_{observed}$ to the standard deviation that would be expected under the hypothesis that animals only use a single strategy. Under this hypothesis, we fitted the behavior in the last 5 training sessions to a single transition curve with parameters $\epsilon$, $\alpha$ and $s$ (see previous section). We then simulated the behavior of an artificial agent that obeys the transitions governed by this fitted curve: on each trial $n$ in the block, the agent would select the correct choice with probability $a_n$. We determined the block-wise standard deviation in performance of the simulated agent, $s_{simulated}$, repeating this simulation for a total of 100 times to obtain the distribution of this measure (box plots in Fig 2B). The reported $p$ value was the probability that $s_{observed} > s_{simulated}$ according to the bootstrap distribution.

For the visualization of block-by-block parameters of the switching dynamics (Fig 2B), we estimated the lapse rate of each block by maximum-likelihood estimation. We fitted a logistic regression model of the observed choices, $y_n$, of the block against a function of the form $p_n = \epsilon + \frac{1-2\epsilon}{1+exp(-\alpha(n-s))}$, and determined the combination of parameters $\epsilon$, $\alpha$, $s$ that maximizes the likelihood of the observed choices. To visualize the expected mean and standard deviation in performance and lapse rate assuming the mouse uses a single strategy, we simulated the uniform strategy as described above for the same number of blocks as in the experimental sessions. The "expected" error bars are centered at $(E, P)$, where $E$ is the mean lapse rates and $P$ is the mean performance across all simulated blocks. The magnitudes of the horizontal and vertical error bars represent the standard deviations of the lapse rates and performances across all simulated blocks, respectively.

## Block Hidden Markov Model

The blockHMM inference procedure was implemented based on the existing ssm toolbox that was previously developed for a wide range of Bayesian state-space models [66].

We added an implementation to this toolbox by specifying a new set of transition and observation probabilities which specify the blockHMM process. Each observation was defined by three vectors, $\boldsymbol{\alpha}$, $\boldsymbol{s}$ and $\epsilon$ representing the parameters of the sigmoidal transition function for each of the $K$ HMM modes (each vector has dimension $K$ x 1). The vectors were initialized to $\boldsymbol{\alpha_i}$ = 4, $s_i$ = 0.2, $\epsilon_i$ = 0.3 for all $1 \leq i \leq K$.

Given the hidden state in block $i$, i.e. given $z_i = k$, the joint log likelihood of the observed choices in the block is defined via the sigmoidal transition function specified by parameters $\alpha_k$, $s_k$, $\epsilon_k$

$$\sigma_k(t) = \frac{1 - 2\epsilon_k}{1 + exp(\alpha_k(t - s_k))} + \epsilon_k \qquad (M.2)$$

where $t$ = 1, 2, . . ., T enumerates the position of the trials in the block.

The log-likelihood for a "signed" choice $y_t$ (the product of choice $c_t$ and hidden state $u_t$) is that of a Bernoulli random variable with a rate of $\sigma_k(t)$.

$$log\, L\, (y_t | \alpha_k, s_k, \epsilon_k) = y_t \log \sigma_k(t) + (1 - y_t)\log(1 - \sigma_k(t)) \qquad (M.3)$$

The joint log-likelihood of the observed choices in the block $i$ is the sum of the log-likelihoods of individual trials

$$log\, P\, (\boldsymbol{y}|z_i = k) = \sum_{t=1}^{T} log\, L\, (y_t | \alpha_k, s_k, \epsilon_k) \qquad (M.4)$$

The joint log-likelihood for the whole session is the sum of the log-likelihood in individual blocks. The hidden states evolved according to a Markovian process with stationary transition governed by a transition matrix $T$ with dimension $K$ x $K$.

The blockHMM was fit with an Expectation-Maximization (EM) algorithm. The hidden states were initialized based on $k$-means clustering with $K$ clusters. The implementation of the EM algorithm was the same as described previously for the ssm toolbox. We used the L-BFGS algorithm for the M-step when updating the values of $\boldsymbol{\alpha}$, $\boldsymbol{s}$ and $\epsilon$, with constraints $\boldsymbol{s} \geq 0.01$, $\boldsymbol{\alpha} \geq 0.01$, $0.01 \leq \epsilon \leq 0.5$.

To evaluate the cross-validated log-likelihood (Fig 3D), we split the data into 80% training set and 20% test set. The blockHMM was run on the training set and the log-likelihood $L_{test}$

was evaluated on the test set. We normalized this cross validated log-likelihood by

$$L_{norm} = \frac{L_{test} - L_0}{n_{test} \, log(2)} \tag{M.5}$$

where $L_0$ is the cross-validated log-likelihood of a null model (a Bernoulli($p$) model where $p$ is the observed fraction of trials where $y_t = 1$), $n_{test}$ is the number of trials in the test set.

## Synthetic agent simulation

The synthetic agent (Fig 3C–3G) was simulated with $K = 3$ HMM modes with parameters $s_1 = 4$, $\alpha_1 = 0.2$, $\epsilon_1 = 0.3$; $s_2 = 1$, $\alpha_2 = 0.8$, $\epsilon_2 = 0.15$; $s_3 = 9$, $\alpha_3 = 1.5$, $\epsilon_3 = 0.05$. The true transition matrix of the agent was

$$T = \begin{bmatrix} 0.966 & 0.003 & 0.031 \\ 0.007 & 0.954 & 0.039 \\ 0.025 & 0.020 & 0.955 \end{bmatrix} \tag{M.6}$$

The behavior was generated for $N = 1000$ blocks, each block consisting of 30 trials.

## BlockHMM fitting to animal behavior

For each animal, we concatenated the behavioral choices from all training sessions into a $B$ x $T$ matrix where $B$ is the total number of blocks from all the sessions, and in each block, we considered the first $T = 15$ choices of the animal. The blockHMM fitting procedure was run on this matrix for $K = 1, 2, 3, \ldots, 6$ modes. We chose the value of $K$ that maximized the normalized log-likelihood of the test set ($L_{norm}$; S2A Fig).

After $K$ is determined, we ran the blockHMM fitting procedure with 3,000 iterations, starting from five different random seeds and selecting the run that results in the highest cross-validated log-likelihood. After fitting the blockHMM model, we recovered parameters $s_k$, $\alpha_k$, $\epsilon_k$ for individual modes in the model. We determined the foraging efficiency $E_k$ by numerically integrating the area under the curve of the choice transition function (with a step size of 0.1)

$$E_k = \int_1^{25} \sigma_k(t) \mathrm{dt} \tag{M.7}$$

Based on the histogram of the mode efficiencies (Fig 4B), which is multi-modal, we defined cut-offs for dividing the modes into three regimes: low performance (below 65%), intermediate performance (65–84%) and high performance (above 84%). Based on the distribution of lapse and offset (Fig 4C), the intermediate-performance group was further split into two subgroups: one with high lapse (lapse rate > 0.1), and one with high offset (lapse rate ≤ 0.1). Each mode in the behavior of the animals was classified into one of these four groups based on the above criteria.

## Simulations of Q-learning and inference-based behavior

*Forward simulations.* We simulated an artificial environment that alternates between two states, "left" and "right", in blocks of trials. The first block was chosen at random to be in the "left" or "right" state, and the state identity flipped for each subsequent block. At the start of each block, we determined the number of trials in the blocks, $N$, by sampling an integer at random in the range [15,25]. We then simulated $N$ trials in the block. In each trial, the agent selected an action and received feedback from the environment. If the block was in the "left" state, left actions yielded reward with probability 100% and right actions yielded reward with

probability of 0%. Conversely, if the block was in the "right" state, left actions yielded reward with probability of 0% and right actions yielded reward with probability of 100%.

Each Q-learning agent was defined by two parameters, the learning rate $\gamma$ and exploration rate $\epsilon$. For our simulations, we simulated a 25 x 20 grid of parameters within the range $0.01 \leq \gamma \leq 1.4$, and $0.01 \leq \epsilon \leq 0.5$. On each trial, the Q-learning agent implemented a Q-value update and selected actions with an $\epsilon$-greedy policy. The agent maintained two values associated with the two actions, $q_L$ for left actions and $q_R$ for right actions. We initialized $q_L = q_R = 0.5$. On each trial, the agent updated these values according to

$$q_i \leftarrow q_i + \gamma(r - q_i) \tag{M.8}$$

where $r$ is the feedback of the trial ($r = 1$ for rewarded actions and $r = 0$ for non-rewarded actions). The Q-learner chose the higher-valued action with probability $1 - \varepsilon$, and selected actions at random (with probability 50% for each choice) on a small fraction $\varepsilon$ of trials.

Each inference-based agent held an internal model which consisted of two hidden states, $L$ and $R$, that corresponded to the unobserved hidden states, "left" or "right", of the environment. The internal model was defined by two parameters, $P_{switch}$ and $P_{rew}$ according to

$$P(s_{i+1} = R | s_i = L) = P(s_{i+1} = L | s_i = R) = P_{switch} \tag{M.9}$$

$$P(s_{i+1} = L | s_i = L) = P(s_{i+1} = R | s_i = R) = 1 - P_{switch} \tag{M.10}$$

$$P(r_i = 1 | s_i = L, c_i = L) = P(r_i = 1 | s_i = R, c_i = R) = P_{rew} \tag{M.11}$$

$$P(r_i = 1 | s_i = L, c_i = R) = P(r_i = 1 | s_i = R, c_i = L) = 1 - P_{rew} \tag{M.12}$$

where $s_i$ refers to the hidden state on trial i and $c_i$ refers to the choice on trial i.

That is, the evolution of the hidden states followed a Markov process with probability $P_{switch}$ of switching states and $1 - P_{switch}$ for remaining in the same state on each trial. For our simulations, we simulated a 15 x 10 grid of parameters within the range $0.01 \leq P_{switch} \leq 0.45$, and $0.55 \leq P_{rew} \leq 0.99$.

We derived a recursive update for the agent's posterior belief about the current world state, given previous choices and feedback. Let $P_L(t) = (s_t = L | c_1, r_1, c_2, r_2, \ldots, c_{t-1}, r_{t-1})$ and $P_R(t) = (s_t = R | c_1, r_1, c_2, r_2, \ldots, c_{t-1}, r_{t-1})$. Then

$$P_L(t) = \frac{1}{\Omega} \sum_{i=L,R} P_i(t-1) \, P(r_{t-1} | s_{t-1} = i) \, P(s_t = L | s_{t-1} = i) \tag{M.13}$$

$$P_R(t) = \frac{1}{\Omega} \sum_{i=L,R} P_i(t-1) \, P(r_{t-1} | s_{t-1} = i) \, P(s_t = R | s_{t-1} = i) \tag{M.14}$$

where $\Omega$ is a normalization factor to ensure $P_L(t) + P_R(t) = 1$.

We initialized $P_L(0) = P_R(0) = 0.5$. On each trial, the agent selected the left action if $P_L(t) > 0.5$, the right action if $P_L(t) < 0.5$, and acted randomly otherwise.

We simulated an ensemble of Q-learning and inference-based agents with parameters as described above. For each agent, the behavior was simulated for a total of 1000 blocks. Parameters of transition dynamics, lapse rate $\epsilon$, slope $\alpha$ and offset $s$, were determined similarly to rodent behavior (see section *Analysis of animal performance* above).

*Backward inference.* The above fitting procedure was performed for all 650 agents (25 x 20 Q-learning and 15 x 10 inference-based agents). To cluster the behavior of artificial agents into groups with qualitatively distinct transition curves, we pooled the four behavioral features, $\epsilon$,

$\alpha$, $s$, and $E$, from these agents to form a 4 x 650 feature matrix, representing 4 features per agent and 650 agents. We applied a density-based clustering method to segment the cloud of points into distinct domains (S6 Fig). First, the four-dimensional features were embedded into a two-dimensional t-SNE space using a Euclidean distance metric. 2-D histograms of the data points in the t-SNE space were formed, and a watershed algorithm was run on the resulting heat map to identify the clusters of high density in this point cloud. We found six distinct clusters using this method. The identities of these clusters were then mapped back to the location in the Q-learning or inference-based parameter spaces.

To validate whether these six classes were meaningful classifications of the artificial agents, a $k$-nearest neighbor (kNN) decoder was trained to predict the behavioral class of agents (Q1-4, IB5-6) based on the observed behavioral transitions. We first generated a synthetic dataset used for training the classifier. We simulated the behavior of each Q-learning and inference-based agent in 50 synthetic experimental sessions with 20 block transitions per session. For each synthetic session, we obtained four parameters of the observed behavior, lapse $\epsilon$, slope $\alpha$, offset $s$, and efficiency $E$. The kNN was trained to predict the class identity (Q1-4, or IB5-6) given these inputs. To balance the number of training examples for different classes in the data set, we subsampled within each class so that each class contains the same number of examples. We split this data into a training set (containing 80% of the data) and a test set (20% of the data). The kNN decoder was fitted on the training set and evaluated on the test set. The accuracy of the decoder was measured both by the fraction of correctly labeled examples per regime, and by the Matthews Correlation Coefficient.

### Decoding the algorithm underlying the blockHMM transition modes

As described above, each mode in the blockHMM was defined by three parameters $s_k$, $\alpha_k$, $\epsilon_k$. The foraging efficiency $E_k$ was also determined for each mode (see *BlockHMM fitting to animal behavior*). Together, the four parameters $s_k$, $\alpha_k$, $\epsilon_k$, $E_k$ were given as inputs into the kNN decoder that was trained in the previous section to infer the behavioral regime (Q1-4, IB5-6) of each of the HMM modes.

### Identification of early switches

To investigate whether some blockHMM modes might correspond to negative offsets ($s < 0$), we included choices of the animal on three trials prior to each block reversal (S9 Fig). This results in a $B$ x $T$ matrix where $B$ is the total number of blocks from all the sessions, and $T = 18$ choices per block (3 choices prior to reversal + 15 choices after the reversal). The number of modes, $K$, was determined again for each animal using the cross-validation procedure described in "blockHMM fitting to animal behavior". Using the selected value of $K$, the blockHMM fitting procedure was carried out on the $B$ x $T$ matrix as described in "blockHMM fitting to animal behavior", yielding parameters $s_{fit}$, $\alpha$, $\varepsilon$, with $s_{fit} > 0$ being the offset with respect to $t = -3$ trials prior to block reversal. The true offset $s$ (with respect to $t = 0$) was determined by subtracting 3 from $s_{fit}$.

In S9A and S9B Fig, the behavioral regimes "low-performing", "intermediate-performing, high-lapse", "intermediate-performing, high-offset", "high-performing" are defined using the same criteria and thresholds as specified in "blockHMM fitting to animal behavior".

### Supporting information

**S1 Fig. Quantification of sources of the sub-optimal performance compared to the win-stay-lose-switch strategy.** (a) Quantification of late performance of all mice across all sessions (mean ± s.e.m, n = 21 mice). Late performance was calculated by averaging each mouse's

performance in the last 10 trials of each block on each session. (b) Quantification of number of initial errors for all mice across all sessions (mean ± s.e.m, n = 21 mice). Dashed lines in (a) and (b) indicate optimal win-stay-lose-switch strategy (WSLS). WSLS strategy should yield 100% late performance, and should only incur 1 initial error per block. (c) Individual animal average performance on the last 5 sessions (shown on y-axis) and average left–right bias on the last 5 sessions (shown on x-axis). Each point represents one animal with the annotated ID. Points are colored by the number of block HMM modes of each animal (Figs 4E and 6A). (d) Difference in performance between left and right blocks (mean ± standard deviation) across all animals on each training session. (e) Performance across training sessions (mean ± standard deviation) for male (n = 9; black) and female mice (n = 12; blue). (f) Performance of male and female mice (mean ± standard deviation) across three session groups: sessions 1–10, sessions 11–20, and sessions 21–30.
(DOCX)

**S2 Fig. Quantification of sex differences in the HMM modes and performance-based behavioral regimes.** (a) Selection of the number of modes, K, used for blockHMM fitting. Each panel shows the normalized cross-validated log-likelihood for different values of K ranging from 1 to 6. The value of K that maximized the cross-validated log-likelihood is indicated by the vertical dashed line. (b) Distribution of the number of blockHMM modes in male and female mice. (c) Composition of performance-based behavioral regimes in male and female mice. Both female and male mice used four behavioral regimes during their learning.
(DOCX)

**S3 Fig. BlockHMM transition matrices for individual experimental animals.** In each matrix, entry $M_{ij}$ shows the transition probability between states $P(z_t = j \mid z_{t-1} = i)$. Each state is labeled with the decoded behavioral strategy (Q1-Q4 or IB5-6) shown in Fig 6A.
(DOCX)

**S4 Fig. Variations of four key behavioral benchmarks across the Q-learning parameter space.** a) Illustration of the sigmoidal transition function with four parameters: switch offset s, switch slope $\alpha$, lapse $\varepsilon$, and overall foraging efficiency E. b) Behavioral features for Q-learning agents in a 100–0 environment. We simulated the behavior of 25 x 20 Q-learning agents with different values of the learning rate $\gamma$ and exploration parameter $\varepsilon$, and measured the four behavioral features for each agent by fitting the average transition function over 1000 blocks to a sigmoidal function. c) Example behavior of three Q-learning agents with a fixed $\varepsilon$ = 0.1 and varying learning rate $\gamma$. Top row shows the behavior of each agent over 100 blocks (each row represents the outcomes of all the trials within a single block, red: incorrect choice, blue: correct choice). Bottom row shows the average transition function (black curve, mean ± standard deviation, n = 1000 blocks), and the fitted sigmoid (blue curve). d) Same as c, but for three Q-learning agents with fixed $\gamma$ = 1.2 and varying $\varepsilon$.
(DOCX)

**S5 Fig. Variations of four behavioral benchmarks across the inference-based parameter space.** a) Map of variations in offset s, slope $\alpha$, lapse $\varepsilon$ and efficiency E, for inference-based agents in the parameter space. The simulations were performed on an ensemble of 15 x 10 inference-based agents with different values of the internal model parameters $P_{rew}$ and $P_{switch}$. b) Example behavior of three inference-based agents taken from the diagonal of the parameter space (dashed line, agents are represented by crosses in panel a plots), illustrating the performance over 100 blocks (top row), and average transition function (bottom row, mean ± standard deviation, n = 1000 blocks).
(DOCX)

**S6 Fig. Subdomains of model-free and inference-based behavior.** Approach to segment the model-free and inference-based parameter spaces into distinct behavioral regimes. We performed a computational simulation of an ensemble of Q-learning and inference-based agents taken from grids that spanned the entire two spaces. For each agent, we obtained the transition function and four behavioral features characterizing the sigmoidal fit. We pooled the features of all agents into a feature matrix and applied a density-based approach to cluster these features into six regimes. The regime identities were visualized for all points in the two parameter spaces. After clustering the points, we identified four regimes in the Q-learning space (Q1-4) and two regimes (IB5-6) in the inference-based space.
(DOCX)

**S7 Fig. Evolution of behavioral strategies of individual mice.** Stacked bar plots in each panel show the composition of the behavioral strategies (Q1-Q4, IB5-IB6) of a single animal over the course of training. Black line on the panel shows the animal's performance in each session.
(DOCX)

**S8 Fig. Sex differences in composition of blockHMM modes.** (a) Average frequency (mean ± standard deviation) of decoded blockHMM behavioral strategies (Q1-Q4, IB5-IB6) across training sessions for female (top panel) and male mice (bottom panel). (b) Mean composition of behavioral strategies in male and female mice across three session groups: sessions 1–10 (top), sessions 11–20 (middle), and sessions 21–30 (bottom panel). (n.s: $p > 0.05$, Mann-Whitney U-test).
(DOCX)

**S9 Fig. Identification of behavioral modes corresponding to early switches.** (a) Distribution of blockHMM mode performances across all experimental animals. BlockHMM modes are classified into low, intermediate and high-performing groups as in Fig 4B. (b) Lapse and offset parameters of individual behavioral modes across all animals. Modes are classified as low-performing (blue), intermediate-performing, high-lapse (yellow), intermediate-performing, high-offset (pink), high-performing (green). A subset of high-performing modes with negative offset are labeled as "Early" switches (black circles). The rest of the high-performing modes are labeled as "Regular" switches. (c) Block transition dynamics of the behavioral regimes identified in (b). Red lines in the bottom panels correspond to the High-performing modes with negative offsets. (d) Frequency of each class of blockHMM modes over the course of training.
(DOCX)

## Acknowledgments

The authors thank Tzuhsuan Ma, Morteza Sarafyazd, John Tauber, Indie Garwood and members of the Sur lab for insightful feedback on the project conceptualization and implementation of the Hidden Markov Model.

## Author Contributions

**Conceptualization:** Nhat Minh Le, Murat Yildirim, Mehrdad Jazayeri, Mriganka Sur.

**Formal analysis:** Nhat Minh Le.

**Funding acquisition:** Mriganka Sur.

**Investigation:** Nhat Minh Le, Murat Yildirim, Yizhi Wang, Hiroki Sugihara.

**Methodology:** Nhat Minh Le, Murat Yildirim.

**Resources:** Nhat Minh Le, Murat Yildirim.

**Software:** Nhat Minh Le.

**Supervision:** Mehrdad Jazayeri, Mriganka Sur.

**Visualization:** Nhat Minh Le.

**Writing – original draft:** Nhat Minh Le.

**Writing – review & editing:** Nhat Minh Le, Murat Yildirim, Yizhi Wang, Hiroki Sugihara, Mehrdad Jazayeri, Mriganka Sur.

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
