## [Decision Letter · Decision Letter 0]

26 Apr 2023

Dear Prof Sur,

Thank you very much for submitting your manuscript "A state-space algorithm for dissociating mixtures of strategies during reversal learning" for consideration at PLOS Computational Biology. As with all papers reviewed by the journal, your manuscript was reviewed by members of the editorial board and by several independent reviewers. The reviewers appreciated the attention to an important topic. Based on the reviews, we are likely to accept this manuscript for publication, providing that you modify the manuscript according to the review recommendations.

Sincerely,

Alireza Soltani

Academic Editor

PLOS Computational Biology

Marieke van Vugt

Section Editor

PLOS Computational Biology

Reviewer's Responses to Questions

**Comments to the Authors:**

Reviewer #1: This is an interesting study that builds on recent state-based mixture models to describe heterogeneity in rodent reward-based decision making behaviors. The study is well-written and the methodology is sound and can be clearly understood (and the data and code are made freely available for closer inspection). The main conclusions are (1) reward-guided decision making behavior in rodents are composed of a diversity of distinct decision strategies -- *even* in very simple all-or-none reversal learning tasks, and *even* after extensive task exposure in such tasks, and (2) these distinct strategies can be described by model-free and inference-based evaluation models (and mice for the most part transition between these strategies throughout early and late phases of training).

My only major concerns are:

(1) conclusion 1 above is more or less expected (we know that mouse behavior is often suboptimal in nearly all task settings, and that state-space models can capture these suboptimalities is already published). Relatedly, the adaptation of these previously published state-space models here seems quite modest. The authors may want to temper/qualify (or perhaps, fortify/better justify, if authors disagree) claims of novelty around the HMM mixture model itself (i.e. in the abstract, title, less emphasis on the state-space model versus the conculsions drawn for it in combination with the model agent simulations)

(2) Relatedly, the adapted state-space model here seems of somewhat limited utility outside of this study. I imagine the utility of a new model should be a) it's ability to describe behavior in new depth (i think the authors adequately meet this bar, although caveat for point 1 above at the high level), and b) providing new parameters for linking reward-based decisions with their underlying neural correlates. Because the model here describes decisions in blocks of choices, it seems unlikely to be very useful for the latter (perhaps it provides a toehold into the neural basis of block transitions?). In addition, this model is only suitable to describe reversal learning tasks with clear block structures. My main suggestion here: can the authors apply their blockHMM to a seperate dataset from a similar related revesal learning task (perhaps datasets from the Intl Brain Lab?). I imagine this could support their main conclusions and demonstrate a broader utility of the model. If this is not possible or feasible, I would ask to better fortify the more general utility of this modeling framework (compared to prior efforts) with regards to a) understanding the neural basis of mixed strategies and b) behavior in related but distinct contexts.

Minor points/questions:

(1) Fig 3e-- can the authors add the actual values of state transitions rather than colorscale to better evaluate

(2) Fig4 -- is the blockHMM fit to each individual animals or concatenaged data? I would be curious to see the cross-validated log-likliehood for different numbers of states in the data (related to Supp Fig 2a), as well as the transition matrices for individual mice or the group.

Reviewer #2: Review is uploaded as an attachment.

**Have the authors made all data and (if applicable) computational code underlying the findings in their manuscript fully available?**

Reviewer #1: Yes

Reviewer #2: Yes

PLOS authors have the option to publish the peer review history of their article (what does this mean?). If published, this will include your full peer review and any attached files.

Reviewer #1: **Yes: **Scott Bolkan

Reviewer #2: No

Figure Files:

Data Requirements:

Reproducibility:

References:

---

## [Decision Letter · Decision Letter 1]

9 Aug 2023

Dear Prof Sur,

We are pleased to inform you that your manuscript 'Mixtures of strategies underlie rodent behavior during reversal learning' has been provisionally accepted for publication in PLOS Computational Biology.

Best regards,

Alireza Soltani

Academic Editor

PLOS Computational Biology

Marieke van Vugt

Section Editor

PLOS Computational Biology

Reviewer's Responses to Questions

**Comments to the Authors:**

Reviewer #1: I thank and congratulate the authors for a thoughtful reply to all comments.

Reviewer #2: The authors have addressed all of my comments more than adequately, and I especially appreciate all the supplementary analyses (and figures) that have been conducted in response to my questions. I feel that the paper has been strengthened after the revision. I hope that this study will appeal to wider audiences.

**Have the authors made all data and (if applicable) computational code underlying the findings in their manuscript fully available?**

Reviewer #1: Yes

Reviewer #2: Yes

PLOS authors have the option to publish the peer review history of their article (what does this mean?). If published, this will include your full peer review and any attached files.

Reviewer #1: **Yes: **Scott S. Bolkan

Reviewer #2: No

---

## [Editor Report · Acceptance letter]

25 Aug 2023

PCOMPBIOL-D-23-00531R1 

Mixtures of strategies underlie rodent behavior during reversal learning

Dear Dr Sur,

I am pleased to inform you that your manuscript has been formally accepted for publication in PLOS Computational Biology. Your manuscript is now with our production department and you will be notified of the publication date in due course.

With kind regards,

Zsofi Zombor
